# Green light triggered [2+2] cycloaddition of halochromic styrylquinoxaline—controlling photoreactivity by pH

Kubra Kalayci[1,2], Hendrik Frisch [1,2✉], Vinh X. Truong [1,2✉] & Christopher Barner-Kowollik [1,2✉]

Photochemical reactions are a powerful tool in (bio)materials design due to the spatial and temporal control light can provide. To extend their applications in biological setting, the use of low-energy, long wavelength light with high penetration propertiesis required. Further regulation of the photochemical process by additional stimuli, such as pH, will open the door for construction of highly regulated systems in nanotechnology- and biology-driven applications. Here we report the green light induced [2+2] cycloaddition of a halochromic system based on a styrylquinoxaline moiety, which allows for its photo-reactivity to be switched on and off by adjusting the pH of the system. Critically, the [2+2] photocycloaddition can be activated by green light (λ up to 550 nm), which is the longest wavelength employed to date in catalyst-free photocycloadditions in solution. Importantly, the pH-dependence of the photo-reactivity was mapped by constant photon action plots. The action plots further indicate that the choice of solvent strongly impacts the system's photo-reactivity. Indeed, higher conversion and longer activation wavelengths were observed in water compared to acetonitrile under identical reaction conditions. The wider applicability of the system was demonstrated in the crosslinking of an 8-arm PEG to form hydrogels (ca. 1 cm in thickness) with a range of mechanical properties and pH responsiveness, highlighting the potential of the system in materials science.

[1] Centre for Materials Science, Queensland University of Technology, 2 George Street, Brisbane, QLD 4000, Australia. [2] School of Chemistry and Physics, Queensland University of Technology, 2 George Street, Brisbane, QLD 4000, Australia. ✉email: h.frisch@qut.edu.au; vx.truong@qut.edu.au; christopher.barnerkowollik@qut.edu.au

Light-gated reactions are important tools in soft matter materials design and the biological sciences due to their unique spatial and temporal control over the chemical processes[1–5]. Critically, by switching the wavelength of irradiation in photo-regulated processes, it is possible to trigger specific reactions in a sequence as well as establish true wavelength orthogonality[6–14]. This distinct feature has become a key concept in the design of photo-responsive systems[15]. To date, most wavelength-dependent reactions require activation by UV light, which has low penetration characteristics and can be harmful to species in biological environments[16]. Thus, identifying a catalyst-free system that does not require (harsh) UV light and can afford deeper light penetration is an ongoing quest. To meet this demand, several synthetic strategies have been developed to red-shift the activation wavelength of photochemical reactions[17]. For example, extending the aromatic conjugation or substituting the electron-donating/withdrawing groups on the chromophores can shift the absorbance and activation wavelength into visible regimes[18].

Further control over photochemical reactions with additional stimuli such as pH, temperature or catalysts provides the opportunity to build highly regulated systems in materials science[9,19–21]. Such complex systems are very beneficial in e.g. drug delivery, where site-specificity and orthogonal reactivity are often required[22]. Furthermore, incorporating a pH switch into a photoreactive system as an additional feature to lock/unlock its photo-reactivity, coupled with long wavelength visible light activation, offers fascinating possibilities for the construction of molecular switches in nanotechnology-based applications[23], and the assembly of chiral functional supramolecular architectures. For example, the enantiodifferentiating [4+4] photocycloaddition of 2-anthracenecarboxylic can be controlled by complexation with a template, via hydrogen-bonding, and the formation of specific enantiomers can be accelerated by tuning pH of the environment[24–27]. In biochemistry, the photo-behaviors of amino acids, such as racemization and decomposition, is strongly dependent on the pH of the environment and this feature has been utilized in the enantioenrichment of biologically important molecules, such as meteorite born amino acids[28,29].

Amongst photochemical reactions, [2+2] cycloadditions have key advantages such as λ-orthogonal reversibility, wavelength-dependent reactivity and lack of any additives[30–35]. Combining these advantages with the ease of functionalizing photoreactive groups, several chromophores with redshifted reactivity (up to 470 nm) such as styrylpyrene[30,32] and acrylamidylpyrene[36] have been introduced by our group and their applications in soft matter design are being actively exploited[30,36,37]. Notably, the acrylamidylpyrene was found to have the most redshifted photoreactivity (470 nm) reported to date for a solution-based [2+2] photocyloaddition. However, the poor solubility of these reactive groups due to the conjugated bulk pyrene unit makes their application challenging at times, especially for biology-related systems. Red-shifting of [2+2] photocycloadditions (λ = 530 nm) was reported in the crystalline state, however, the precise design of the crystalline structures was in this case essential to induce the photocycloaddition, which limits its broader applicability[38]. Furthermore, additional features to control the photo-reactivity of the [2+2] cycloaddition system have yet to be successfully implemented. The use of two orthogonal stimuli—wavelength and pH—to finely control the formation of dynamic covalent linkages will enable the development of advanced smart mechanisms for biomaterials engineering.

Herein, we break new ground in catalyst-free photoligation reactions on two levels: (i) by introducing the first green light activated [2+2] photocycloaddition in solution and (ii) by reporting halochromism of a photoreactive unit that translates into its photo-reactivity, allowing to reversibly switch a photo-reaction on and off by varying the pH. The compact structure of the developed styrylquinoxaline (SQ) moiety—compared to pyrene units usually required for visible light activated [2+2] photocycloadditions—significantly enhances the solubility of the compound in aqueous media.

## Results

**Design and [2+2] cycloaddition of styrylquinoxaline.** The judicious design of the molecular architecture is key to accomplishing red-shifted reactivity in [2+2] photocycloadditions. By replacing one phenyl ring of stilbene with quinoxaline, styrylquinoxaline was obtained and a carboxylic acid functionality was introduced as a versatile precursor for the conjugation to polymers (see Supplementary Fig. 17). The chromophore was subsequently attached to a poly(ethylene glycol) (PEG) mono methyl ether via carbodiimide coupling, affording PEG-SQ as shown in Fig. 1. PEG was selected due to its solubility in a wide range of solvents—and its wide application in biomaterials design[30,39]. The absorption of PEG-SQ is significantly red-shifted compared to stilbene as well as all alkene derivatives that can undergo [2+2] photocycloadditions reported to date[9,40,41].

To investigate the photo-reactivity of the styrylquinoxaline moiety, [1]H NMR, UV/vis, SEC and SEC-MS analyses were performed before and after irradiation of PEG-SQ solutions in water (10 mg mL$^{-1}$) at 510 nm—a wavelength that has thus far not been reported to induce photocycloadditions in solution (Fig. 2, also see Supplementary Figs. 5 and 23). After irradiation, the absorbance at longer wavelengths in the UV/vis spectrum decreases, while new absorbance bands appear close to λ = 250 and 325 nm. Such a change in the UV/vis spectrum is well-known to occur upon cyclobutane ring formation between two alkenes, which decreases the extent of the conjugated system (Fig. 2a)[42].

To combine these spectroscopic observations related to the electronic structure changes with the macromolecular ligation

**Fig. 1 Schematic representation of the proposed reaction mechanism.** [2+2] Photocycloaddition of PEG-styrylquinoxaline (PEG-SQ).

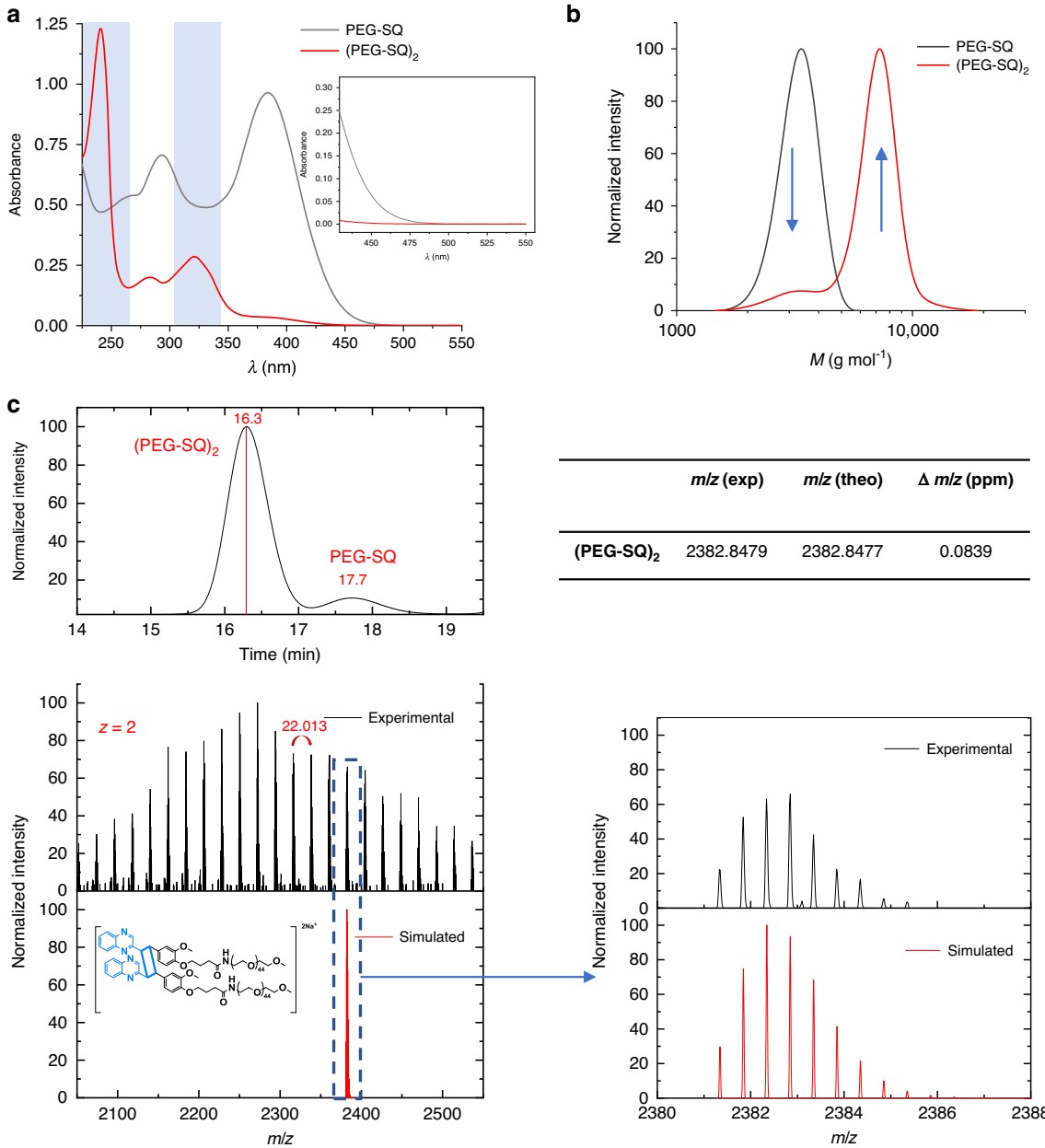

**Fig. 2 Characterization of the cycloaddition product. a** UV/vis spectra of PEG-SQ and (PEG-SQ)$_2$ dimer after irradiation at 510 nm with $9.29 \times 10^{22}$ photons in water (10 mg mL$^{-1}$), with the gray bars showing the appearance of new absorption peaks specific to quinoxaline due to the change in conjugation after irradiation at 510 nm. The insert shows the expanded absorbance spectra between 450 and 550 nm. **b** SEC data (calibrated against PMMA) of PEG-SQ and (PEG-SQ)$_2$ after irradiation at 510 nm with $9.29 \times 10^{22}$ photons in water (10 mg mL$^{-1}$). **c** SEC-MS analysis of cycloaddition of PEG-SQ after irradiation at 510 nm with $9.29 \times 10^{22}$ photons. SEC trace of the (PEG-SQ)$_2$ peak at 16.3 min jointly with the PEG-SQ peak at 17.7 min. High resolution MS spectrum displaying doubly charged species of the (PEG-SQ)$_2$ peak after 16.3 min of elution in the SEC trace. The $m/z$ difference between patterns is 22.013, corresponding to half of one PEG repeating unit. Zoom into the MS spectra of experimental and simulated isotopic patterns, indicating their excellent agreement. The inset table shows the theoretical and experimental $m/z$ values of the (PEG-SQ)$_2$ dimer.

reactions, SEC traces of PEG-SQ before and after irradiation at 510 nm with $9.29 \times 10^{22}$ photons were recorded (Fig. 2b). After irradiation, a distribution ($M_n = 7100$ g mol$^{-1}$) at approximately double the initial molecular weight ($M_n = 3200$ g mol$^{-1}$) can be observed, while the intensity of the initial molecular weight distribution (PEG-SQ) decreased.

To confirm the [2+2] cycloadduct formation on the molecular level, SEC hyphenated to high resolution mass spectrometry was carried out[43]. The SEC trace of PEG-SQ exhibits only one peak at 17.7 min (see Supplementary Fig. 23), while the SEC trace after irradiation displays the dimer peak of PEG-SQ ((PEG-SQ)$_2$)

at 16.3 min –as well as a small amount of unreacted PEG-SQ with a peak at 17.7 min (Fig. 2c). The mass spectrum obtained at 16.3 min of SEC elution time displays multiple patterns of the PEG repeating units with a difference of $m/z = 22.013$, equal to the molecular weight of one PEG repeating unit. The simulated isotopic patterns of (PEG-SQ)$_2$ overlap with the experimentally obtained $m/z$ values, confirming the structure of the photocycloaddition product.

**Wavelength and pH-dependent reactivity studies**. After confirming the [2+2] cycloadduct formation, the wavelength-dependent reactivity was investigated. Since a previous study[36]

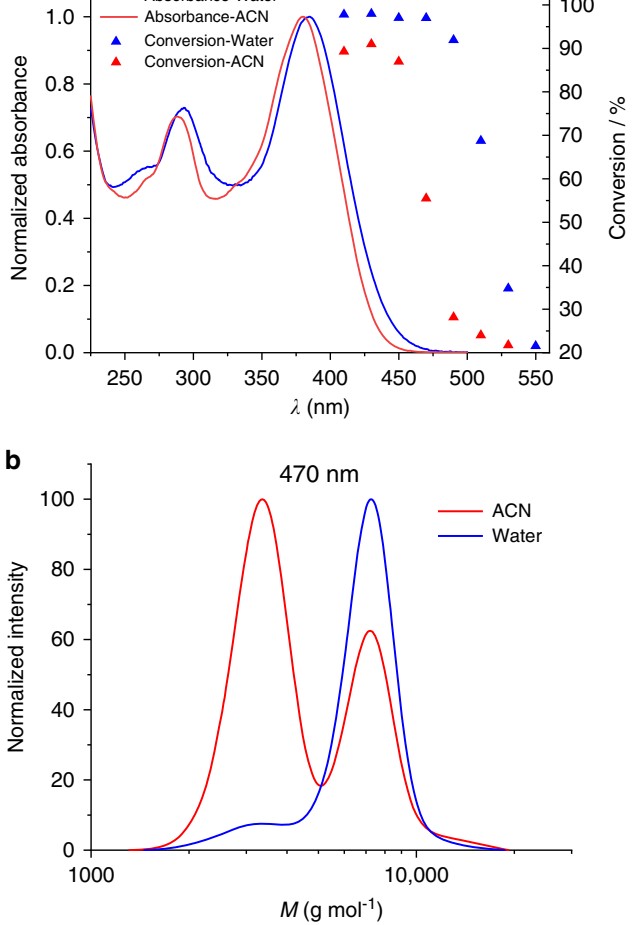

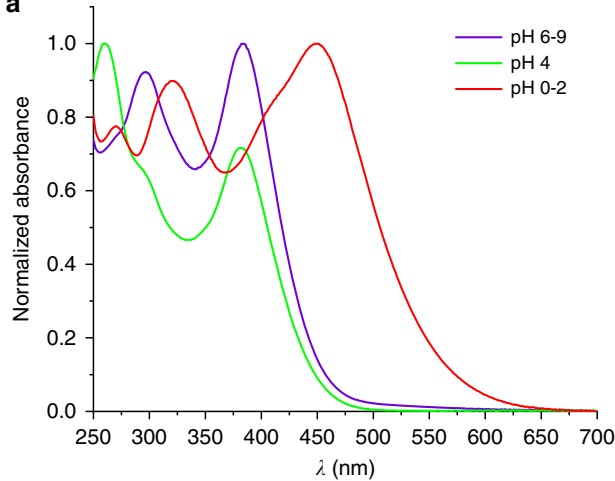

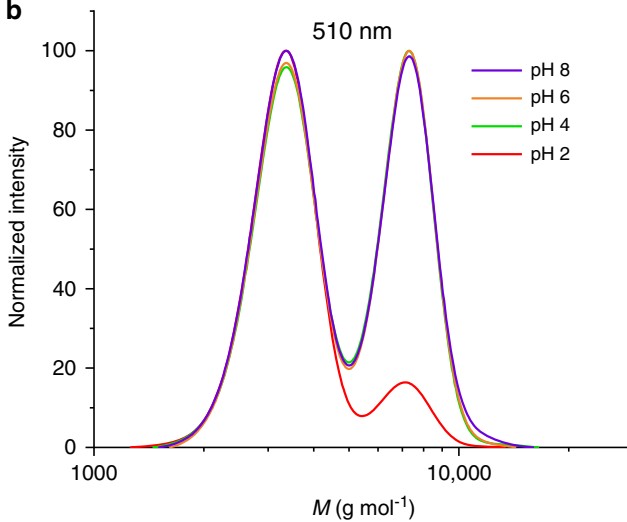

**Fig. 3 Solvent dependent action plots. a** Action plot and UV/vis spectra of PEG-SQ in water and acetonitrile (10 mg mL$^{-1}$ solutions irradiated with $1.23 \times 10^{22}$ photons). **b** SEC traces showing conversion in acetonitrile (55.5%) and water (96.9%) after irradiation at 470 nm with the same number of photons ($1.23 \times 10^{22}$).

**Fig. 4 pH-dependent reactivity studies. a** UV/vis spectra of PEG-SQ solutions at different pH values. **b** SEC data of PEG-SQ solutions (10 mg mL$^{-1}$) in water with different pH after irradiation at 510 nm with the same number of photons ($1.23 \times 10^{22}$).

indicated that solvent effects can alter the reactivity of [2+2] photocycloadditions, the conversion of PEG-SQ to (PEG-SQ)$_2$ was monitored in water and in acetonitrile using the same concentration (10 mg mL$^{-1}$) and number of photons ($1.23 \times 10^{22}$) for each sample. A tunable laser setup was used as a monochromatic light source, and all samples were irradiated with the same number of photons at different wavelengths to establish the wavelength-dependent conversion as a so-called 'action plot' (Fig. 3a). We earlier reported that the wavelength-dependent reactivity does not necessarily align with the absorption spectra of the employed chromophores[32,44,45]. The same trend was observed here for PEG-SQ. Although very little absorption was observed close to 450 nm, the conversion up to 450 nm is close to quantitative in both solvents after irradiation with $1.23 \times 10^{22}$ photons, albeit slightly higher in water (97.0%). Remarkably, above 450 nm the reactivity difference in water and in acetonitrile significantly increases. The conversion in water at 470 nm is very close to its maximum, while the conversion in acetonitrile is close to 55.5%. The SEC data explicitly show the conversion difference in water and in acetonitrile after irradiation at 470 nm with the same number of photons (Fig. 3b, also see Supplementary Fig. 4a, b). These results indicate that solvent interactions affect the photo-reactivity and the activation wavelength of [2+2] cycloaddition reactions.

To investigate the effect of pH on the photo-reactivity of PEG-SQ, solutions with different pH values were prepared. Driven by

the protonation of the quinoxaline nitrogens[46,47] (see Supplementary Fig. 1), the UV/vis spectrum obtained at pH 2 shows a significant bathochromic absorbance shift resulting in a red colored solution (Fig. 4a, also see Supplementary Figs. 2 and 3). Each solution was irradiated with the same wavelength (510 nm) and identical number of photons to observe the change in reactivity. SEC data indicate that at acidic levels (pH 2) the reactivity decreases significantly, whereas it is not affected at higher pH values (pH > 2) (Fig. 4b). As a consequence, the halochromic properties of the styrylquinoxaline unit appear to translate directly into its reactivity, marking it as the first example of a photochemical ligation that can be directly tuned by switching the pH.

**Hydrogel studies.** To demonstrate the utility of our photochemical reaction design in material science, we synthesized an 8-arm (PEG-SQ)$_8$ with a molecular weight of 20,000 g mol$^{-1}$ via amide formation, and examined its crosslinking in aqueous solution (Fig. 5a) under irradiation with light in the same wavelength range (400–510 nm) investigated on linear PEG-SQ. Rheological data show a rapid increase in the storage modulus (G') value upon irradiation with light (Fig. 5b), indicative of the

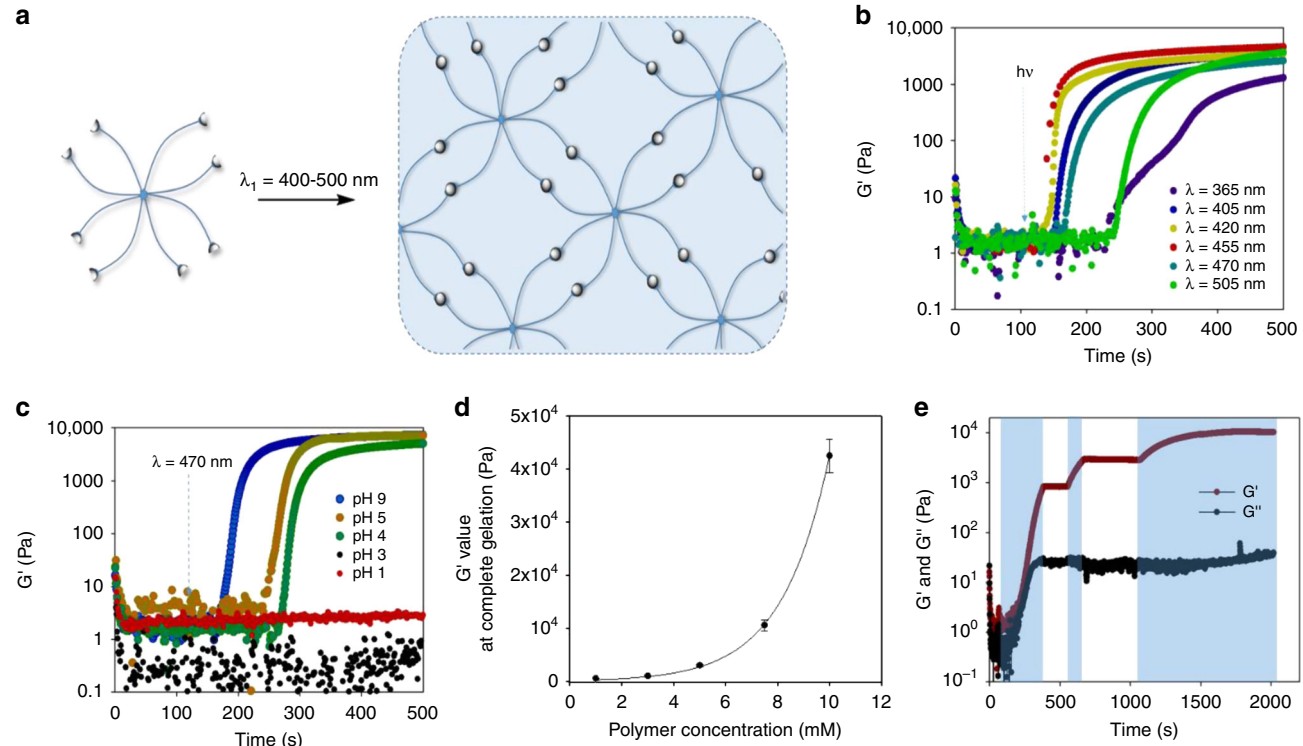

**Fig. 5 Characterization of hydrogels. a** Schematic representation of crosslinking of 8-arm $(PEG\text{-}SQ)_8$ (MW = 20,000 g mol$^{-1}$) via photocycloaddition to form a hydrogel under irradiation of light. Gelation process of a polymer solution ($c = 7.5$ mM) followed by the evolution of the storage modulus G' as a function of time; **b** under irradiation of different wavelengths using an LED light source (WheeLED Mightex); and **c** irradiation at $\lambda = 470$ nm and at different pH values. **d** G' values at quantitative gelation from different polymer concentrations between 1 and 10 mM (error bars represent standard deviation from a mean value of 3 measurements). **e** Temporal control of the gelation process of polymer solution with $c = 7.5$ mM by switching light on and off at different time intervals.

formation of a crosslinked network. Under the same irradiation intensity $I = 20$ mW cm$^{-2}$, the gelation rate is highly dependent on the wavelength of light with $\lambda = 420$ and 455 nm being the most efficient, providing complete gelation in 10 min (see Supplementary Fig. 8). This result is strongly correlated with the action plot data (Fig. 3a), highlighting the importance of the action plot study in understanding the photo-reactivity of the photochemical reaction, confirming that UV/vis absorbance of the chromophore may not be indicative of the reactivity of its associated photochemical reaction system[32,44]. Of note, light at $\lambda = 365$ nm also triggered gelation, however, at a very slow rate. This is due to the competing cycloreversion under UV light irradiation, as suggested by data from linear PEG-SQ, where a sample was irradiated at 470 nm and 360 nm subsequently and the change in molecular weight was monitored via SEC (see Supplementary Fig. 6). After irradiation at 360 nm, an increase in the intensity of the non-dimerized PEG-SQ peak was observed, which is indicative of a competing reverse reaction. Furthermore, acidic pH significantly affected the crosslinking of $(PEG\text{-}SQ)_8$ with pH 4 slowing down the gelation rate and fully suppressing the crosslinking at pH 1 (Fig. 5c). Importantly, the photo-reactivity was restored when pH was switched back to neutral or basic (refer to the Supplementary videos), demonstrating the reversibility of using a pH switch in the photo-induced cycloaddition process.

Taking advantage of the high solubility of the SQ end group, we attempted to crosslink the polymer solutions at different concentration ($c = 1$–10 mM) to obtain gels with a range of physical properties. We observed an exponential increase in the stiffness by increasing the polymer concentration, with the modulus values varying from 0.5 kPa to 40 kPa (Fig. 5d).

These values are similar to the moduli reported in 8-arm PEG-based hydrogels prepared by photocycloaddition-driven crosslinking[48].

Beer-Lambert' law dictates that the rate of photocycloaddition has an inverse exponential relationship with regard to the competitive absorption of the product[49]. As seen from Fig. 2a, the dimer product displayed an absorbance in the 350–450 nm region. Thus, the photocycloaddition process can be significantly affected in thick and static samples due to the competitive absorption of the dimer product. Indeed, when the polymer solution ($c = 5$ mM) having thicknesses of *ca.* 1 cm was irradiated with light at $\lambda = 405$ nm or 420 nm, only a thin layer of gel (thickness ≤0.2 cm) was formed after 1 h of irradiation (Fig. 6). More efficient gelation across the entire sample depth was observed when the polymer solution was irradiated with light at $\lambda \geq 455$ nm, and a gel with thickness of 1 cm was formed under green light ($\lambda = 455$) illumination. This is a significant improvement of sample size for hydrogel fabrication via catalyst-free photocycloaddition, compared to a thus far reported thickness of ≤0.1 cm for curing with UV light[50–52] or short wavelength blue light-initiated crosslinking[30,32,36]. This result emphasizes the advantage of longer wavelength activation of photochemical system with regard to penetration depth and circumventing competitive absorption of the resultant photo-adduct.

To investigate the suitability of the introduced visible light-triggered coupling in biomaterials engineering, we attempted to crosslink 8-arm $(PEG\text{-}SQ)_8$ solutions ($c = 5$ mM) containing mouse fibroblasts (L929 cell line) under irradiation of green light at $\lambda = 505$ nm and $I = 20$ mW cm$^{-2}$ (Fig. 7). Live/dead staining of the cell-laden hydrogels showed very high cell viability

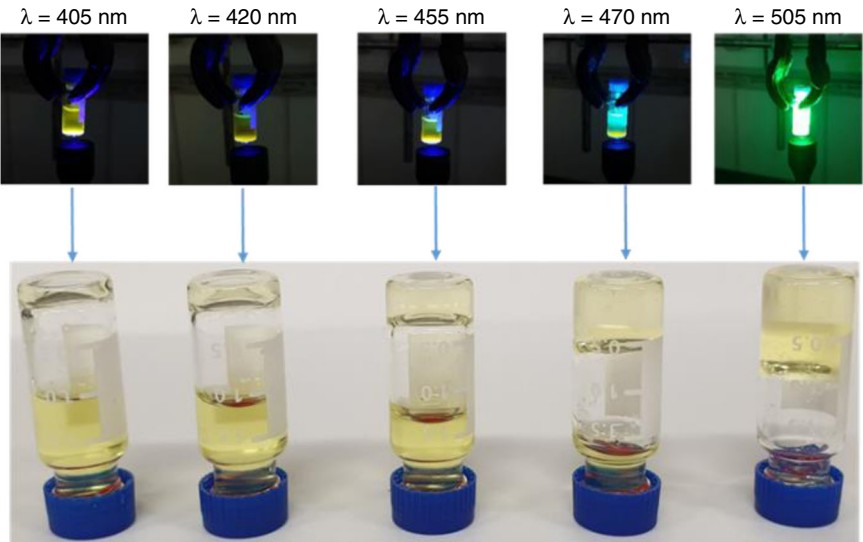

**Fig. 6 Effect of light penetration depth on hydrogel thickness.** Photographic images of gels formed under irradiation at different wavelengths ($\lambda = 405$–$505$ nm, $I = 20$ mW cm$^{-2}$) from a polymer solution ($c = 5$ mM) with thickness of 1 cm.

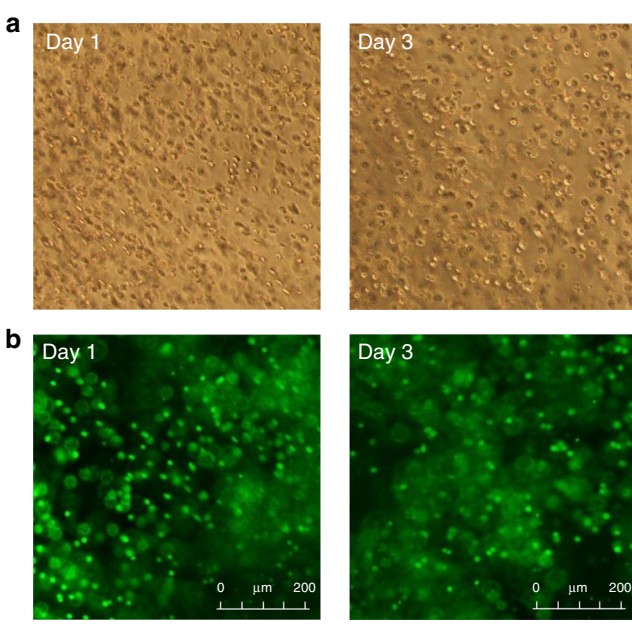

**Fig. 7 Cell viability studies. a** Bright-field images of hydrogels containing encapsulated fibroblasts (L929); and **b** live/dead staining of the cell-laden hydrogels after 1 day and 3 days of 3D culture, showing cell viability ≥80% (live cell = green, dead cell = red).

(*ca.* 90%) after 1 day of encapsulation, indicating that the crosslinking process is non-toxic to fibroblasts. Notably, the cells remained viable (viability = 80%) after 3 days of 3D culture inside the hydrogel substrate. This viability is high considering PEG is non-fouling toward cells in general, and high rates of fibroblasts cell death have been reported when they are encapsulated in PEG-based hydrogels for more than 24 h[53,54].

## Discussion

We introduce a halochromic species combining photo-induced [2+2] cycloadditions with a pH switch to exercise fine control over photo-reactivity. With this system, it is possible to trigger a [2+2] photocycloaddition with green light ($\lambda < 550$ nm)—the longest activation wavelength so far employed in catalyst-free photocycloaddition. The wavelength-dependent reactivity was monitored in different solvents, revealing different wavelength-dependent reactivities and highlighting the importance of interactions between solvent and chromophore. Furthermore, we evidence that the photo-reactivity can be tuned by the pH, particularly, at acidic pH it is possible to switch the reactivity off. Finally, we demonstrate the applicability of the system by the selective formation of hydrogels with different mechanical properties and explicitly show that green light allows for higher penetration depths, enabling the fabrication of thicker hydrogels. We submit that applications of the developed photoligation system can be readily extended to 3D laser lithography, drug delivery, and pharmacology. In the context of cell biology, our system can potentially be employed in cell-related studies, where pH-dependent cellular responses need to be mapped. Furthermore, the utility of initiating a photoligation in a higher wavelength regime and simultaneously tuning its reactivity with an orthogonal pH switch is a critical step toward molecular surgery, altering selected parts of (macro)molecules while leaving others untouched.

## Methods

**Synthetic procedures**. Detailed synthetic procedures are described in the Supplementary Information and are accompanied with reaction schemes and NMR characterizations figures.

**Dimerization of PEG-SQ and calculation of action plots**. The incident light used for laser experiments was a Coherent Opolette 355 tunable OPO operated at 410–500 nm with a full width half maximum of 7 ns and a repetition rate of 20 Hz. The emitted pulse, which has a flat-top spatial profile, was expanded to 6 mm diameter using focusing lenses and directed upwards using a prism. The beam was then centered on a glass laser vial which is positioned in a 6 mm diameter slot in a temperature-controlled sample holder. The energy transmitted through the sample holder was measured using a Coherent Energy Max PC power meter.

**PEG-SQ**. (1 mg) was dissolved in 0.1 mL of Acetonitrile or water in laser vials, the vials were crimped airtight and degassed for 5 min applying Argon. Each solution was irradiated at $\lambda = 410$–$530$ nm with $1.23 \times 10^{22}$ photons. The conversion of the dimerization reactions was calculated from the SEC data (Supplementary Fig. 4) by calculating the integral of the peaks. Detailed information on the photochemical procedures as well as the scheme of laser set-up can be found in the Supplementary Information section.

**Rheology studies**. Rheological experiments were studied using an Anton Paar Physica rheometer with a plate-plate configuration. The lower plate is made of

quartz and the upper plate is made of stainless steel with a diameter of 15 mm. A liquid light guild, which was connected to the WheeLED light source, was equipped below the quartz plate. In a typical experiment, 50 μL of a solution of **P2** was placed on the lower plate and the upper plate was brought to a measurement gap of 0.2 mm. A layer of paraffin oil was applied on the edge of the stainless-steel plate to prevent dehydration of hydrogel and the test was started by applying a 1% strain with the frequency of 0.1 Hz on the sample.

**Hydrogel swelling**. To study the effect of pH on hydrogel swelling, a polymer solution ($c = 7.5$ mM, 200 μL) in a 5 mL sealed vial was irradiated with light at 455 nm for 3 h. The resultant solid gel was extracted from the vial and placed in excess PBS solution pH 7.4. The weight of the hydrogel was monitored until no further change in the weight was observed. The pH of the solution was subsequently adjusted using HCl 1 M and NaOH 1 M solution. At each set pH, the weight of the hydrogel ($w_t$) was recorded when no further change in the weight was observed. The calculation of swelling ratio was demonstrated in Supplementary Information section.

**Cell culture study**. Cell culture was carried out using commercial mouse fibroblasts L929 (NCTC clone 929, ATCC® CCL-1™). Cells were cultured on tissue culture flasks as per manufacturers' instructions, then trypsinised with Tryple Express to detach from the culture surfaces. The cells were centrifuged for 3 min at 0.3 g and the supernatant discarded. For cell culture studies, fibroblasts were resuspended in PEG-(SQ)$_8$ solution ($c = 5$ mM) to achieve a cell density of $5 \times 10^6$ cells per mL. The solution was agitated gently to allow the cells to distribute throughout the solution and pipetted into tissue culture inserts. The inserts were exposed to green light ($\lambda = 505$ nm, $I = 20$ mW cm$^{-2}$) irradiation for 30 min and cell culture media (Dulbecco's Modified Eagle Medium) were added. Triplicates were prepared. The cell-laden hydrogel samples were rinsed twice with culture media and maintained at 37 °C and 5% CO$_2$. At day 1 cell culture media were exchanged once. To assess cell viability, gels were removed from tissue culture inserts after day 1 and day 3 in culture, washed in PBS and stained using Live/Dead® Viability/Cytotoxicity Kit for mammalian cells (Invitrogen) following the manufacturer's recommended protocol.

## Data availability

The authors declare that the main data supporting the findings of this study are available within the article and its Supplementary Information files. Extra data are available from the corresponding author upon request.

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

## Acknowledgements

C.B.-K. acknowledges the Australian Research Council (ARC) for funding in the context of a Laureate Fellowship enabling his photochemical research program, an ARC Discovery Grant focused on red-shifting photochemical reaction systems as well as continued key support from the Queensland University of Technology (QUT). The size exclusion chromatography hyphenated with mass spectrometry data reported herein were obtained at the Central Analytical Research Facility (CARF) operated by the Institute for Future Environments (IFE).

## Author contributions

All authors contributed to discussion and evaluation of the results at all stages. K.K., H.F., and V.X.T. conceived and designed the experiments. H.F., V.X.T., and C.B.-K. motivated and supervised the research project. K.K. and V.X.T. performed the experiments and prepared all figures. K.K. and V.X.T. wrote the paper in close collaboration with H.F. and C.B.-K.

## Competing interests

The authors declare no competing interests.
