## [Peer Review File · Nature Communications]

REVIEWER COMMENTS

Reviewer #1 (Remarks to the Author):

This manuscript describes a pH tunable cycloaddition reaction driven by low energy photonic input (green light 550 nm). It presents studies on a model linear polymer and then extends to a dendrimer case that leads to cross linked gels. The mechanics of the gels could be tuned by dendrimer load and the optical density could be optimized to allow for penetration of the photostimulus at centimeter distances through the macroscale material. Several beneficial properties were noted in the context of biomaterials applications, and a proof of principle fibroblast live dead assay was performed.

A major claim meant to justify a Nature level publication is the lowest energy wavelength for cycloaddition under catalyst free conditions. This may very well be, and seems to be an important advance. I did not mine the literature to corroborate this, and I trust the authors did their due diligence on this point. However, what is needed for this report is to assess comparative performance with the known art. For example, what are the penetration depths for the prior catalyst free photodimerization gel materials? Is there a marked difference in live-dead outcomes with this lower energy excitation vs the higher energy precedents? Are the mechanical properties here substantial deviations from past work? Although the aspect of lower energy excitation is important for the prospective biomedical applications, if the present work provides no significant suggestion for dramatic improvement in functional properties, then the work to me would seem more incremental and better suited for a specialized macromolecular or materials journal.

The work claims the first example of a photochemical ligation that can be directly tuned by switching the pH. The authors should better defend this claim through analysis of prior art. Many examples of carboxylic acid terminated reactants in similar photochemical reactions are known, for example where these molecules are templated via H bonds to a template molecule, and in some cases the templation interactions are enhanced by the pH of the solution (for example allowing carboxylates to interact with the H-bond donor template more strongly than the latent carboxylic acids). The work here seems unique but should be prefaced more specifically with prior art.

Do they know that the nitrogen ring is doubly protonated? This seems like it would require a much stronger acid, not simply pH 2. It seems that the same argument could be made for a singly protonated quinoxaline ring leading to the same effect.

Reviewer #2 (Remarks to the Author):

This manuscript comprises work on a photo-induced [2+2] cycloaddition of two stilbene-type molecules (being integrated into a PEG polymer). The key is using a stilbene unit, which contains a 2-quinoxaline substituent and a PEG-bound benzene ring (SQ). This allows performing photo reactions at wavelengths in the visible region. The authors show that the absorption spectrum of the title compound is pH dependent and that the cycloaddition is photo-reversible at 360 nm. This can be

translated into the photo triggered formation of a gel (using PEG-(SQ)₈) with additional pH dependence.

Overall, the findings reported here are innovative and suitable for Nature Comm.; however some details should be addressed by the authors:

a) In Figure 1a, the abs spectra between 450 and 550 nm should be shown at an expanded scale (in terms of abs) as an insert. It is intriguing that the absorption spectra of PEG-SQ and that ascribed to (PEG-SQ)₂ are very much alike. Do the authors have an explanation why the compound with a smaller chromophore has almost the same UV/Vis spectrum as the conjugated stilbene derivative or is the red spectrum just an overlay between the spectrum of PEG-SQ and one band of (PEG-SQ)₂, at ca. 240 nm with a sh at ca 310 nm? This, then, would also indicate a rather low conversion (quantum efficiency) in terms of the cycloaddition. In this view, the authors should also consider the photo-induced trans/cis conversion of the 'stilbene moiety'.

b) It would be desirable if the SEC-determined dimerization efficiency could be connected to the photo-chemical observations.

c) The action plots show a clear but not drastic influence of the solvent, accordingly, the statement 'that solvent interactions play a critical role' is somehow overstated.

d) In my view, the same holds for the following section: 'SEC data indicate that at acidic levels (pH 2) the reactivity decreases significantly, whereas it is not affected at higher pH values (pH >2) (Figure 3b). As a consequence, the halochromic properties of the styrylquinoxaline unit appear to translate directly into its reactivity, marking it as the first example of a photochemical ligation that can be directly tuned by switching the pH.' It is straightforward, that the penetration depth at the wavelength of 510 nm used for the corresponding experiments is different since it is connected with the corresponding absorbance (at 510 nm), which is shifted by a solvatochromic/halochromic effect. It should be clearly stated if this is (just) a consequence of the higher absorptivity/lower penetration rather than 'other reactivity'. Experiments at lower concentrations of the system should provide some evidence here.

Reviewer #3 (Remarks to the Author):

The major claims of the paper are the long-wavelength absorption properties of the stilbene-type photoactive systems and the pH-dependency of the photodimerization as well as the formation of high-molecular weight products from polymeric substrates that also influences the formation of supramolecular structures such as hydrogels. This is of course a step forward that applies research on photocycloadditions of styrenes and stilbenes that has been in the focus of photochemistry research over the last 4-5 decades and applications in material sciences of these reactions are numerous. The novel aspect here is that one part of the chromophore that allows strong red-shift is on the other hand also applicable as a proton-induced switch for photoreactivity, i.e. induces a halochromic effect. This is a remarkable and obviously useful effect that allows to switch on and off the photoreactivity depending on local pH.

This effect might be important for cellular experiments where pH differences might be mapped by the efficiency of photocycloaddition and thus compartmental properties might become sensible by

this reaction. This would go obviously beyond the idea of this paper but should be shortly elaborated.

Overall, the single ideas behind the specific photochemistry are not completely novel. But the combination of halochromic features and substituent effects that allow red-shift together with the molecular weight increase that allows supramolecular aggregation effects looks impressive and appears to me as an important feature that will influence the thinking in the fields of classic photochemistry and supramolecular photochemistry.

Concerning the data and the experimental information given in the paper, all results are sound and appear very well studied and reported. I recommend publication of this important novel combination of several features of photochemistry and supra-molecular chemistry that also influences the field of modern material chemistry.

Reviewer #1

This manuscript describes a pH tunable cycloaddition reaction driven by low energy photonic input (green light 550 nm). It presents studies on a model linear polymer and then extends to a dendrimer case that leads to cross linked gels. The mechanics of the gels could be tuned by dendrimer load and the optical density could be optimized to allow for penetration of the photostimulus at centimeter distances through the macroscale material. Several beneficial properties were noted in the context of biomaterials applications, and a proof of principle fibroblast live dead assay was performed.

Comment:

A major claim meant to justify a Nature level publication is the lowest energy wavelength for cycloaddition under catalyst free conditions. This may very well be and seems to be an important advance. I did not mine the literature to corroborate this, and I trust the authors did their due diligence on this point.

Response: We have extended our literature review to photocycloadditions in the crystalline state and included a study where [2+2] cycloadditions were performed in the crystalline state induced by green light under catalyst free conditions (DOI: 10.1021/jacs.6b11857) and mentioned this in the introduction. We have modified the associated sentences in the main text.

Comments:

However, what is needed for this report is to assess comparative performance with the known art.

- a) For example, what are the penetration depths for the prior catalyst free photodimerization gel materials?**

Response: The reviewer raises the valid point that the comparison to the state of the art was not fully developed. Our photochemical crosslinking allows for the fabrication of hydrogels with thickness ≥ 1 cm. This is a significant improvement from general thin film of gels (≤ 0.1 cm) prepared using UV light- or short wavelength blue light-initiated photocycloadditions. We have added this comparison and the relevant references (ref 47-49) to the manuscript.

- b) Is there a marked difference in live-dead outcomes with this lower energy excitation vs the higher energy precedents?**

Response: In general, long wavelength UV light-initiated crosslinking allows for 3D cell encapsulation, however, the cells are typically not viable for more than 24 h post-encapsulation, e.g. significant cell death was reported for cells trapped in PEG hydrogels prepared by blue light crosslinking (ref 37). In our system, we observed cell viability for up to 3 days after encapsulation, refer to the cell study in the results and discussion, specifically Figure 6. This observation suggests that the longer-wavelength activated crosslinking is highly suitable for 3D cell cultures.

- c) Are the mechanical properties here substantial deviations from past work? Although the aspect of lower energy excitation is important for the prospective biomedical applications, if the present work provides no significant suggestion for dramatic**

improvement in functional properties, then the work to me would seem more incremental and better suited for a specialized macromolecular or materials journal.

Response: The storage moduli of hydrogels prepared in our work are similar to values reported based on 8-arm PEG hydrogels prepared by UV light-initiated photocycloaddition (ref 46). However, the longer wavelength activation and pH-control reactivity constitute the key advances of the current work, critically increasing the utility of [2+2] photocycloaddition in (bio)materials design. We have highlighted the potential of our photochemical systems in the Conclusion in more depth in response to the reviewer's query.

Comment:

The work claims the first example of a photochemical ligation that can be directly tuned by switching the pH. The authors should better defend this claim through analysis of prior art. Many examples of carboxylic acid terminated reactants in similar photochemical reactions are known, for example where these molecules are templated via H bonds to a template molecule, and in some cases the template interactions are enhanced by the pH of the solution (for example allowing carboxylates to interact with the H-bond donor template more strongly than the latent carboxylic acids). The work here seems unique but should be prefaced more specifically with prior art.

Response: We thank the reviewer for their helpful comment. In response, we have now in detail explored pH dependent photochemical ligations in the introduction and have placed them more appropriately into the context of the current study. In the same context, we have added additional references.

Comment:

Do they know that the nitrogen ring is doubly protonated? This seems like it would require a much stronger acid, not simply pH 2. It seems that the same argument could be made for a singly protonated quinoxaline ring leading to the same effect.

Response: We thank the reviewer for pointing this matter out. Based on a further literature search on the protonation of quinoxaline derivatives (DOI: 10.1039/c8cc02018c and DOI: 10.1039/c6cc06443d), we carried out a small test experiment by adding an excess amount of concentrated sulfuric acid to a quinoxaline solution. We observed a colour change from yellow to red and then blue (Figure S1, Supporting Information). Therefore, we believe that the red colour may be an indication of the mono-protonated state, whereas the blue colour can be the result of the doubly protonated state. We have added additional data from this experiment to the Supporting Information section (Figure S1 and S2) and changed the protonated structure in Scheme 1 and within the ToC figure.

Reviewer #2

This manuscript comprises work on a photo-induced [2+2] cycloaddition of two stilbene-type molecules (being integrated into a PEG polymer). The key is using a stilbene unit, which contains a 2-quinoxaline substituent and a PEG-bound benzene ring (SQ). This allows performing photo reactions at wavelengths in the visible region. The authors show that the absorption spectrum of the title compound is pH dependent and that the cycloaddition is photo-reversible at 360 nm. This can be translated into the photo triggered formation of a gel (using PEG-(SQ)₈) with additional pH dependence. Overall, the findings reported here

are innovative and suitable for Nature Comm.; however some details should be addressed by the authors:

Comments:

a) In Figure 1a, the abs spectra between 450 and 550 nm should be shown at an expanded scale (in terms of abs) as an insert. It is intriguing that the absorption spectra of PEG-SQ and that ascribed to (PEG-SQ)₂ are very much alike. Do the authors have an explanation why the compound with a smaller chromophore has almost the same UV/Vis spectrum as the conjugated stilbene derivative or is the red spectrum just an overlay between the spectrum of PEG-SQ and one band of (PEG-SQ)₂, at ca. 240 nm with a sh at ca 310 nm? This, then, would also indicate a rather low conversion (quantum efficiency) in terms of the cycloaddition. In this view, the authors should also consider the photo-induced trans/cis conversion of the 'stilbene moiety'.

Response: We thank the reviewer for the constructive feedback. We have inserted expanded absorption spectra between 450 and 550 nm to the Figure 1a. We have also carried out additional photoreactivity experiments to obtain the maximum conversion of the PEG-SQ to PEG-SQ dimer by increasing the number of photons ($9.29 \cdot 10^{22}$ photons) and concentration (5 mg mL⁻¹). Figure 1a and 1b now include the new results. However, we have not been able to completely separate the (PEG-SQ)₂ dimer from the small amount of unreacted PEG-SQ. Therefore, the UV/vis spectrum indicated as (PEG-SQ)₂ contains an overlay of a small amount of PEG-SQ and (PEG-SQ)₂. Nevertheless, with the new data, we were able to show a significant change in the absorption spectra before and after irradiation of PEG-SQ. In addition, we have performed irradiation experiments with very low concentrations ($2.5 \cdot 10^{-3}$ mg mL⁻¹) to investigate the cis/trans isomerisation of the chromophore. However, we could not observe any significant change in the UV/vis spectrum. We have included the results in the Supporting Information section (Figure S6, Supporting Information).

b) It would be desirable if the SEC-determined dimerization efficiency could be connected to the photo-chemical observations.

Response: As indicated in the above response, we have replaced the SEC and UV/vis figure (Figure 1a and 1b) with the new data. We submit that the decrease in the absorption around 400 nm as a result of PEG-SQ to (PEG-SQ)₂ dimer conversion can correlate with the conversion of PEG-SQ peak to (PEG-SQ)₂ dimer peak in the SEC data.

c) The action plots show a clear but not drastic influence of the solvent, accordingly, the statement 'that solvent interactions play a critical role' is somehow overstated.

Response: We agree that the statement may sound overstated and have removed the word 'critical' from the sentence.

d) In my view, the same holds for the following section: 'SEC data indicate that at acidic levels (pH 2) the reactivity decreases significantly, whereas it is not affected at higher pH values (pH >2) (Figure 3b). As a consequence, the halochromic properties of the styrylquinoxaline unit appear to translate directly into its reactivity, marking it as the first example of a photochemical ligation that can be directly tuned by switching the pH.' It is straightforward, that the penetration depth at the wavelength

of 510 nm used for the corresponding experiments is different since it is connected with the corresponding absorbance (at 510 nm), which is shifted by a solvatochromic/halochromic effect. It should be clearly stated if this is (just) a consequence of the higher absorptivity/lower penetration rather than 'other reactivity'. Experiments at lower concentrations of the system should provide some evidence here.

Response: The reviewer raises the interesting question if the decreased reactivity at 510 nm at pH 2 results predominantly from the red shifted absorption maximum that decreases the light penetration depth at this wavelength. While the previous action plot studies cited in the manuscript demonstrate that reactivity and absorption do not necessarily align, the wavelength dependent action plot in water shows the highest conversions (>90%) at the wavelength of the highest absorbance. Even though the initial penetration depth of light is - as noted by the reviewer - very low, it increases with an increasing number of photons, since the cycloadduct has a smaller conjugated system that leads to a blue shifted absorption. With proceeding irradiation time, the absorbance at longer wavelengths decreases, allowing for the required light penetration depth. Since the absorbance at 410 nm at a neutral pH is not significantly lower than the absorbance at 510 nm at pH 2, the difference in conversion cannot be attributed to light penetration alone (compare Fig 2b and 3b).

Experiments at lower concentrations are very intriguing, however, the quantum yields of [2+2] photocycloadditions are generally concentration dependent, which would further decrease the reactivity.

Reviewer #3

The major claims of the paper are the long-wavelength absorption properties of the stilbene-type photoactive systems and the pH-dependency of the photodimerization as well as the formation of high-molecular weight products from polymeric substrates that also influences the formation of supramolecular structures as hydrogels. This is of course a step forward that applies research on photocycloadditions of styrenes and stilbenes that has been in the focus of photochemistry research over the last 4-5 decades and applications in material sciences of these reactions are numerous. The novel aspect here is that one part of the chromophore that allows strong red-shift is on the other hand also applicable as a proton-induced switch for photoreactivity, i.e. induces a halochromic effect. This is a remarkable and obviously useful effect that allows to switch on and off the photoreactivity depending on local pH.

This effect might be important for cellular experiments where pH differences might be mapped by the efficiency of photocycloaddition and thus compartmental properties might become sensible by this reaction. This would go obviously beyond the idea of this paper but should be shortly elaborated.

Overall, the single ideas behind the specific photochemistry are not completely novel. But the combination of halochromic features and substituent effects that allow red-shift together with the molecular weight increase that allows supramolecular aggregation effects looks impressive and appears to me as an important feature that will influence the thinking in the fields of classic photochemistry and supramolecular photochemistry. Concerning the data and the experimental information given in the paper, all results sound and appear very well studied and reported. I recommend publication of this important novel combination of several features of photochemistry and supra-molecular chemistry that also influences the field of modern material chemistry.

Response: We sincerely thank the reviewer for their valuable remarks. We have added a sentence to the discussion addressing the utility of system's pH dependence in the context of cell-related applications.

REVIEWERS' COMMENTS:

Reviewer #1 (Remarks to the Author):

The authors have satisfied my initial concerns, and I thank them for digging deeper into the prior literature to frame this exciting work in the best light. I only have one lingering request, which is for the authors to specifically state what the current "record" is in terms of low-energy excitation to trigger the dimerization in solution, to the best of their knowledge/awareness. They mention crystalline example at 530 nm, but it would be good to put the solution value here of 550 nm into context of where other solution studies have pushed to. Is the cited 470 nm reported in their prior work considered the low energy number to beat, or has this been pushed longer by other groups/chromophores?

Reviewer #2 (Remarks to the Author):

The authors have thoroughly addressed the the comments/suggestions of all reviewers. I suggest accepting the manuscript in its revised form.

RESPONSE TO REVIEWERS' COMMENTS

Reviewer #1

The authors have satisfied my initial concerns, and I thank them for digging deeper into the prior literature to frame this exciting work in the best light. I only have one lingering request, which is for the authors to specifically state what the current "record" is in terms of low-energy excitation to trigger the dimerization in solution, to the best of their knowledge/awareness. They mention crystalline example at 530 nm, but it would be good to put the solution value here of 550 nm into context of where other solution studies have pushed to. Is the cited 470 nm reported in their prior work considered the low energy number to beat, or has this been pushed longer by other groups/chromophores?

Response: We thank the reviewer for their valuable feedback. To the best of our knowledge, the current record for a [2+2] cycloaddition in solution is 470 nm which was reported in our previous paper. We have added a sentence in the introduction part to highlight the current record.

Reviewer #2

The authors have thoroughly addressed the comments/suggestions of all reviewers. I suggest accepting the manuscript in its revised form.

Response: We thank the reviewer for their positive assessment.